# Detection of Necrosis in Digitised Whole-Slide Images for Better Grading of Canine Soft-Tissue Sarcomas Using Machine-Learning

**DOI:** 10.3390/vetsci10010045

**Published:** 2023-01-08

**Authors:** Ambra Morisi, Taran Rai, Nicholas J. Bacon, Spencer A. Thomas, Miroslaw Bober, Kevin Wells, Michael J. Dark, Tawfik Aboellail, Barbara Bacci, Roberto M. La Ragione

**Affiliations:** 1Department of Comparative Biomedical Sciences, School of Veterinary Medicine, University of Surrey, Guildford GU2 7AL, UK; 2Centre for Vision Speech and Signal Processing, University of Surrey, Guildford GU2 7XH, UK; 3AURA Veterinary, Guildford GU2 7AJ, UK; 4National Physical Laboratory, Teddington TW11 0LW, UK; 5Department of Comparative, Diagnostic, and Population Medicine, College of Veterinary Medicine, University of Florida, Gainesville, FL 32610, USA; 6Department of Microbiology, Immunology and Pathology, Colorado State University, Fort Collins, CO 80523, USA; 7Department of Veterinary Medical Sciences, University of Bologna, 40126 Bologna, Italy; 8Department of Microbial Sciences, School of Biosciences, University of Surrey, Guildford GU2 7XH, UK

**Keywords:** canine, digital pathology, grading, machine learning, necrosis detection, soft-tissue sarcoma

## Abstract

**Simple Summary:**

Canine soft-tissue sarcomas are a group of tumours that arise from the skin and subcutaneous connective tissue. The most common method used to predict the behaviour of these tumours is grading. The grading system used for soft-tissue sarcomas is derived from a combined score calculated by evaluating the mitotic count, percentage of tumour necrosis and degree of cellular differentiation. However, these parameters are highly subjective and a high inter-observer variability has been reported in grading these tumours, which can result in complications regarding treatment plans. Manual identification of areas of necrosis is a time-consuming task that is prone to observer error. Artificial-intelligence algorithms and, in particular, machine learning, can help improve grading by automatically detecting regions of necrosis. The aim of this study was to differentiate image regions in order to automatically identify tumour necrosis in digitised canine soft-tissue sarcoma slides. This method showed an accuracy of 92.7% which represents the number of correctly classified data instances over the total number of data instances. Therefore, the proposed method is a promising tool to minimise human error in the evaluation of necrosis in soft-tissue sarcomas, and hence increase the efficiency and accuracy of histopathological grading of canine soft-tissue sarcomas.

**Abstract:**

The definitive diagnosis of canine soft-tissue sarcomas (STSs) is based on histological assessment of formalin-fixed tissues. Assessment of parameters, such as degree of differentiation, necrosis score and mitotic score, give rise to a final tumour grade, which is important in determining prognosis and subsequent treatment modalities. However, grading discrepancies are reported to occur in human and canine STSs, which can result in complications regarding treatment plans. The introduction of digital pathology has the potential to help improve STS grading via automated determination of the presence and extent of necrosis. The detected necrotic regions can be factored in the grading scheme or excluded before analysing the remaining tissue. Here we describe a method to detect tumour necrosis in histopathological whole-slide images (WSIs) of STSs using machine learning. Annotated areas of necrosis were extracted from WSIs and the patches containing necrotic tissue fed into a pre-trained DenseNet161 convolutional neural network (CNN) for training, testing and validation. The proposed CNN architecture reported favourable results, with an overall validation accuracy of 92.7% for necrosis detection which represents the number of correctly classified data instances over the total number of data instances. The proposed method, when vigorously validated represents a promising tool to assist pathologists in evaluating necrosis in canine STS tumours, by increasing efficiency, accuracy and reducing inter-rater variation.

## 1. Introduction

Soft-tissue sarcomas (STSs) are tumours derived from mesenchymal tissues [1,2,3,4]. In dogs, they develop most frequently in the subcutis where they represent between 9 and 15% of all cutaneous or subcutaneous tumours [3,5,6]. Histological assessment of canine STSs is traditionally performed by microscopic analysis of tissue sections on glass slides. The most-used histological parameter to prognosticate canine STSs and predict their outcome following surgery is the tumour grade, derived from a combined score calculated based on cellular differentiation, mitotic index and percentage of tumour necrosis [7,8,9,10]. The application of these histologic criteria allows individual STSs to be categorised into three distinct grades (I, low grade; II, intermediate grade or III, high grade) [5,9,10,11,12]. Tumour necrosis is a common feature of solid tumours caused by ischaemic injury, owing to rapid rates of tumour growth [13]. However, manual identification and calculation of necrotic regions by visual inspection can be a time-consuming and error-prone task for large whole-slide images [14] and can lead to inter-observer variability [15]. Recent technological advances, on the other hand, would allow histological tumour slides to be converted into digital image datasets for automated analysis. Utilising machine learning to interrogate patterns in these digital histological images may address some of the limitations of manual grading [16]. Machine-learning algorithms have been evaluated with success in the field of human oncology and histopathology for glioma, renal clear cell carcinoma, breast cancer, gastric carcinoma, prostate cancer, and non-small-cell lung cancer [16,17,18,19,20,21,22,23,24,25]. Machine-learning methods have been pivotal to investigating the degree of necrosis in pathology images. Sharma et al. [14] used machine learning for necrosis detection in gastric carcinomas with the best average cross-validation rate (which estimates how accurately a predictive model will perform in practice) of 85.3%. In a second study published by the same authors [14], the proposed CNN architecture for necrosis detection in human gastric carcinoma had the best overall rate of 81.4%. In a more recent study performed by Arunachalam et al. [26] on human osteosarcomas, the accuracy in detecting areas of necrosis using their proposed model was 92.7%.

In this study, we attempt to differentiate image regions in order to identify tumour necrosis in the haematoxylin and eosin (H&E)-stained WSIs of canine STSs (cSTSs). There were several motivations for this study. We previously published [27] the first report on the use of deep learning to detect cSTSs in haematoxylin and eosin (H&E)-stained whole slides. However, the study reported here builds on the initial study and focuses on grading. Necrosis is a specific determinant in assigning a histological grade to cSTSs, and so it is important to recognise and quantify necrosis in STS sections. More generally, the presence of necrosis is considered a characteristic of malignancy and subjectively influences the pathologist’s judgement on tumour behaviour. Furthermore, there are no specific histological stains for necrosis which makes automatic detection of necrosis a highly desirable objective. Automatic necrosis detection could decrease viewing times for pathologists, reduce inter-observer variabilities and hence increase the accuracy of diagnosis and prognosis. Lastly, these methods could also be applied to other histopathology datasets. In this study we applied a pre-trained DenseNet161 CNN model to automatically detect necrosis in cSTSs from WSIs.

## 2. Materials and Methods

### 2.1. Dataset and Slide Annotation

A total of 90 WSIs of canine perivascular-wall tumours (cPWTs), a subtype of cSTSs, were collected from the Department of Microbiology, Immunology and Pathology, Colorado State University, Fort Collins. The digitised slides were reviewed, and the diagnoses and grades were confirmed by two board-certified veterinary pathologists.

Of the 90 cPWTs, necrosis, which was characterised by loss of cellular detail and presence of eosinophilic amorphous material, was identified in 21 cases. The 21 WSIs containing necrosis (one WSI for each case) were manually annotated by a board-certified veterinary pathologist. The same slides were annotated separately by a veterinary surgeon after training in the annotation of necrosis in WSIs. The pathologist had regular follow-up discussions with the veterinary surgeon regarding the annotation procedure and annotation rules. The annotations were made using the open source Automated Slide Analysis Platform (ASAP) software by delineating the contour of necrotic areas using mouse clicks. To avoid potential human bias and increase accuracy, only areas of consensus where the veterinary pathologist and the veterinary surgeon agreed were included in the analysis [16].

### 2.2. Pre-Processing

Patches or “tiles” of 256 × 256 pixel size, were extracted from a large WSI and fed into a pre-trained convolutional neural network. Every patch containing at least 30% necrosis derived from the expert mark-ups was extracted to create the necrosis class. As a proportion of the WSIs did not contain necrosis, and to avoid class imbalance between necrosis versus non-necrosis patches, a subset of all the non-necrosis patches was extracted. It should be noted that these non-necrosis patches were chosen at random. An example of the patch extraction of necrosis is given in Figure 1.

Patch-based approaches are required as whole-slide images are too large to pass through CNN architectures.

The aim of this study was to classify entire patch images; therefore, as patches may contain a mix of necrotic and non-necrotic tissue, a threshold needed to be considered for determining patches containing necrosis (positive patches) and patches that did not contain necrosis (negative patches). Histopathological images with multiple levels of magnification can depict various types of information. For this study, 20× objective magnification was deemed appropriate following a discussion with the pathologists. The higher magnification allows clear identification of necrosis in a WSI image while a lower magnification resulted in loss of the spatial information. However, higher magnifications include sub-cellular regions that are not relevant to the task and may negatively affect the segmentation process.

For the 20× magnification, the threshold applied for negative patch was 0.75 (75% of the patch must not contain necrosis for it to be labelled as non-necrosis). This means that at least 25% of the patch must contain necrosis for it to be labelled as necrosis (positive patch). The total percentage of necrosis present in a slide was calculated from the intersection of the ground truth labelling from human experts. The four grade 3 cSTS slides had 10.92%, 3.05%, 10.06% and 10.59% of necrosis. The grade 2 cSTS slides contained 0.74%, 1.24%, 0.00% and 0.07% of necrosis. None of the grade 1 cSTS slides contained necrosis. These data were used for training, validating and testing the algorithm.

A pre-trained DenseNet161 CNN model was implemented in Python using the PyTorch library and experiments were performed on Dell T630 system, including two Intel Xeon E5 v4 series 8-Core CPUs, four Nvidia Titan X GPUs and 128GB of RAM. 

### 2.3. DenseNet161

Several pre-trained networks were investigated [28,29] and DenseNet161 CNN was deemed appropriate. According to the results reported in a study conducted by Talo [30], DenseNet161 can be used for fast and accurate classification of histopathology images to assist pathologists in their daily clinical tasks. The DenseNet161 model is one of the DenseNet group of models designed to perform image classification [31]. In our deep learning set-up, there were two components: a feature extractor model that was pretrained on the ImageNet dataset [32] and a dense classification layer model. The weights in the convolutional layers of the feature extractor were frozen and the dense classification layer was amended for a binary classification task. Features were extracted using the feature extractor and passed into the dense classification layer model. Training was implemented using the Adam optimiser [33], a learning rate of 0.0001, and a batch size of 32. Test-time normalization was also implemented.

### 2.4. Training, Validation and Testing

The dataset was divided into several subsets as follows: a *training dataset* which is the set of data that are used to train and make the model learn the features/patterns in the data; a *validation dataset* which is a set of data, separate from the training set, that is used to validate the model performance during training; and a *test dataset* which is a separate set of data used to test the model *after* completion of the training. Although the validation and test datasets are similar, test datasets are “unseen” whereas validation is used as an informative dataset during training.

To ensure statistical robustness, 3-fold cross validation was implemented. Three experiments named fold 1, fold 2 and fold 3 were run. The three experiments, fold 1, fold 2 and fold 3, contained 3824, 3754 and 3990 non-overlapping patches of necrosis, respectively, for training, and 1960, 2030 and 1794 patches of necrosis, respectively, for validation (Table 1).

For training, 100 epochs were used. The best model selected was the one with the lowest validation loss.

After training and validation, the algorithm was tested on a test dataset (Table 2).

The test dataset contained 35,202 patches extracted from 12 WSIs (four slides for each tumour grade). The patches that contained necrosis were identified as “positive” and the sum of the patches extracted from the slide containing necrosis and normal tissue were identified as “total”. The total percentage of necrosis present in a slide was then calculated. A probability map for each slide was also generated for a WSI for visualisation purposes.

## 3. Results

The results from the study are presented below in Figure 2, Figure 3 and Figure 4 and Table 3, Table 4 and Table 5. After training the algorithm, the estimated model’s accuracy was computed using 3-fold cross-validation of the dataset. Training accuracy ranged from 89.2% to 98.0%. The training loss ranged from 0.058 to 0.295 (Figure 2). The proposed CNN architecture reported favourable results, with an overall validation accuracy of 92.7% for necrosis detection which represents the number of correctly classified data instances over the total number of data instances (Table 3).

Overall, the validation accuracy ranged from 88.6% to 94.7%. Validation average class-wise accuracy ranged from 89.9% to 94.4%. The validation loss ranged from 0.143 to 0.269 (Figure 3).

After training and validating the algorithm, we computed *accuracy*, *sensitivity*, *precision* and F1 *score* (Table 3) using the number of true positives (TPs), true negatives (TNs), false positives (FPs) and false negatives (FNs).

*Accuracy* represents the number of correctly classified data instances over the total number of data instances that was calculated as follows:$$Accuracy:\;\frac{TP+TN}{TP+TN+FP+FN}$$The results ranged from 89.1% to 92.7%.

*Recall*, also known as *sensitivity* or *true positive rate*, is defined as follows:$$Recall:\;\frac{TP}{TP+FN}$$Recall ranged from 93.4% to 94.6%.

*Precision* (positive predictive value) in classifying the data instances is defined as follows:$$Precision:\frac{TP}{TP+FP}$$Precision ranged from 22.4% to 30.0%.

*F1-score* is a metric, which considers both *precision* and *recall* and is defined as follows:$$F1\text{-}score:\;2*\frac{Precision\times Recall}{Precision+Recall}$$F1-score ranged from 36.2% to 45.4%.

The data in Table 4 represent the test data for the three experiments (folds 1, 2 and 3) with the number of predicted positive patches (patches that contain necrosis), the total number of patches and the percentage of necrosis in each slide. The data in Table 5 illustrate the predicted % of necrosis that the algorithm detected in each slide after training and the % of necrosis annotated by the pathologists.

A tumour necrosis-prediction map for each slide was also generated for a WSI for visualisation purposes (Figure 4). The prediction for true positive, false positive, true negative and false negative are expressed in red, orange, clear and green, respectively.

## 4. Discussion

In this study, we present a detailed report describing the automatic assessment of WSIs for the detection and quantification of necrosis in cSTSs, providing further insight and analysis from our baseline approach as previously published [27]. The experiments presented in this study confirmed that DenseNet161 is able to recognise areas of necrosis with high accuracy (92.7%). Recent studies have shown that CNN can be substantially deeper, more accurate, and more efficient to train than other models [31]. In the current study we used a dense convolutional network (DenseNet). This CNN has several advantages including reducing the number of parameters that need to be learnt in the training phase. At the 0th and close to 0th epoch, the training and validation accuracy were close to 90%. From the experimental datasets it was observed that all three folds produced remarkably consistent results (fold 1 accuracy 92.7%, fold 2 90.8% and fold 3 89.1%). However, the F1 score and precision had lower results (fold 1 F1 score 78.5%, fold 2 75.7% and fold 3 62.9%; fold 1 precision 70.5%, fold 2 63.2% and fold 3 46.8%). This could be due to the fact that this method may fail to generalise to smaller datasets.

On further analysis of the results for necrosis detection, it appears that DenseNet161 had high overall accuracy rates (92.7%) in detecting areas of necrosis in canine STSs from a WSI. This model currently outperforms several methods described in the literature [14,16,26], as we detail below.

In the first study conducted by Sharma et al. [14] the machine-learning model used for necrosis detection in gastric carcinomas had the best average cross-validation rate of 85.3%. In a second study by the same authors in 2017, the proposed CNN architecture for necrosis detection in human gastric carcinoma had the best overall rate of 81.4%. In a more recent study performed by Arunachalam et al. on human osteosarcoma, the accuracy in detecting areas of necrosis using their proposed model was 92.7%, which is identical to the results presented here. Hence, our proposed model DenseNet161 achieved a comparatively favourable performance outcome. These results are unlikely to be influenced by tumour type as the histopathological appearance of coagulative necrosis is the same, however further validation studies should be performed to confirm this. It was also interesting to note that half the number of WSIs were required to train the deep convolutional models compared to similar studies on detection of viable and necrotic tumours in human osteosarcoma [26]. This could be due to the fact that our model is pre-trained and therefore requires less data compared to a non-pre-trained model to achieve similar results.

Furthermore, from the data presented in the learning curves, it can be observed that they are generally constant with a decreasing training error and increasing validation accuracy. It can be also noted that the validation loss is nearly constant due to characteristics of validation data, but training loss, which represents the summation of errors in the model, decreases and validation accuracy increases to become constant. This is a desired characteristic in training as it indicates the model is not overfitting the training data.

As a final validation step, tumour necrosis-prediction maps were generated to display the necrotic regions. These maps can be used to calculate the percentage of tumour necrosis in each patient and visualise the extent of the tumour necrosis over the whole-slide image.

The number of false positive and false negatives are controlled by the threshold, and they can be adjusted as required to suit a specific problem. In our case, we wanted to minimise the number of false negatives. From the data presented here, it appears there is a very low rate of false negatives meaning that very few areas of necrosis are likely to be missed, but that there is a tendency to overestimate areas of detected necrosis (false positives). This can be seen in our results where the algorithm detected more areas of necrosis compared to the ground truth labelling from human experts. However, this overall approach can provide a rapid first pass through a given WSI, which, whilst not intending to replace the skilled expert pathologist, can provide a rapid early indication of areas requiring expert attention.

Analysis of the datasets revealed several limitations to the study as follows: firstly, like all other deep-learning applications published to date in human medicine [14,16,26], our method also requires training with large-scale datasets containing thousands of images. This issue could be addressed using data-augmentation strategies and/or by increasing the number of cases from different sources. Secondly, the ground-truth data were generated by expert pathologists annotating WSIs, which demands a significant time investment from specialists, in common with the vast majority of other supervised deep-learning approaches in clinical sciences. Thirdly, training the proposed model requires approximately four to five days and the availability of significant GPU computer resource. Both issues could be rectified with a large increase in resources, both human and computational, with the additional required financial investment. Lastly, tissue slides can vary in appearance due to the biopsy technique, slide preparation, staining, processing and scanning techniques used in different pathology laboratories [34]. However, our dataset only used slides collected from one institution in order to reduce these variations as much as possible. Future studies should focus on implementing methods for automatic colour and intensity normalisation of WSIs.

There are currently multiple aspects of necrosis assessment that need to be defined. First of all, the current methods for determining percent of tumour necrosis in cSTSs have been poorly defined, meaning that it is difficult for others to replicate, leading to intra- and inter-observer variability [35]. In veterinary medicine, it has not yet been demonstrated whether the percentage of tumour necrosis should be determined grossly (which would have to be confirmed microscopically), histologically or both [35]. The current methods for calculating the percent of tumour necrosis are not well defined [34]. In addition, the number of sections of cSTSs that should be obtained in order to get an accurate representation of necrosis is, as yet, unknown [35]. Roccabianca et al. [7] suggested using one tissue block for each 2 cm diameter. The introduction of standardised trimming would aid in reducing the variability. However, this would require the generation of a large number of sections in order to improve accuracy, and thus would significantly add to the workload of the pathologist. The use of machine-learning algorithms in assessing the percentage of tumour necrosis would allow for a larger number of sections to be assessed with minimal input from the pathologist. This approach would allow a larger number of slides to be evaluated and therefore a more representative sample to be checked without impacting on the time taken by the pathologist to report each case. This approach would also help eliminate the current scenario where technicians may avoid creating sections of tumours in areas where they appear necrotic, haemorrhagic or oedematous when trimming samples [35]. Future studies should investigate these areas by including outcome data.

## 5. Conclusions

In conclusion, DenseNet161 was able to recognise areas of necrosis in cSTSs with high accuracy. This suggests that such an approach could improve the performance of pathologists by offering high sensitivity and reducing inter-observer variability. These results demonstrate that AI can potentially be used as an effective diagnostic support tool to grade cSTSs with more accuracy. Future studies could investigate the accuracy of the prediction maps and correlate the results with patient outcome to better define tumour grade. In addition, researchers could investigate optimising thresholds to improve the sensitivity, specificity, precision and F1-score.

## Figures and Tables

**Figure 1 vetsci-10-00045-f001:**
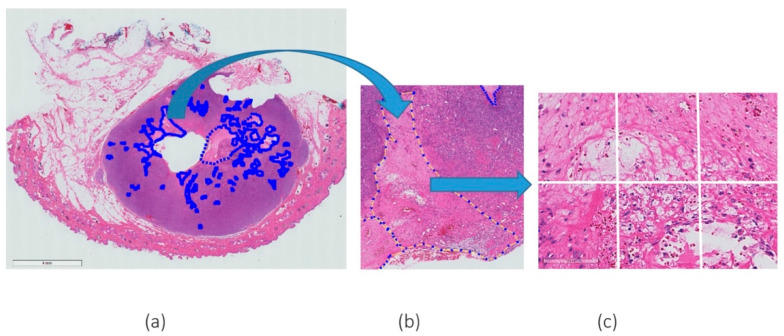
Example of H&E WSI with (**a**) a few annotations of necrosis marked by expert pathologists, (**b**) a magnified (magnification 5×) region of agreement of both pathologists and (**c**) an example of patches extracted from the annotated region (magnification 20×).

**Figure 2 vetsci-10-00045-f002:**
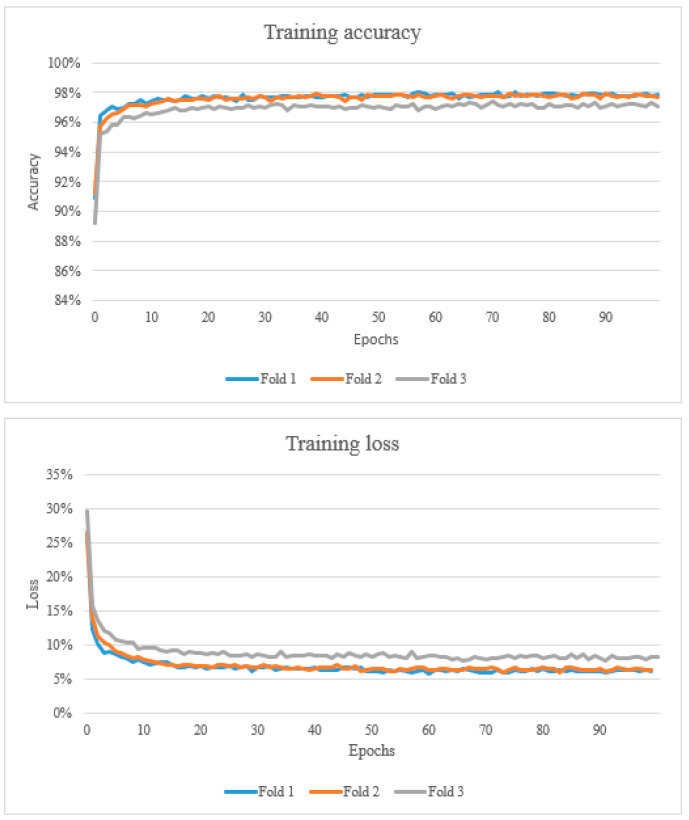
The top graph shows the training accuracy (learning curve). A learning curve is a plot that shows epochs on the x-axis and learning or improvement on the y-axis. The bottom graph shows the training loss which represents the summation of errors in the model.

**Figure 3 vetsci-10-00045-f003:**
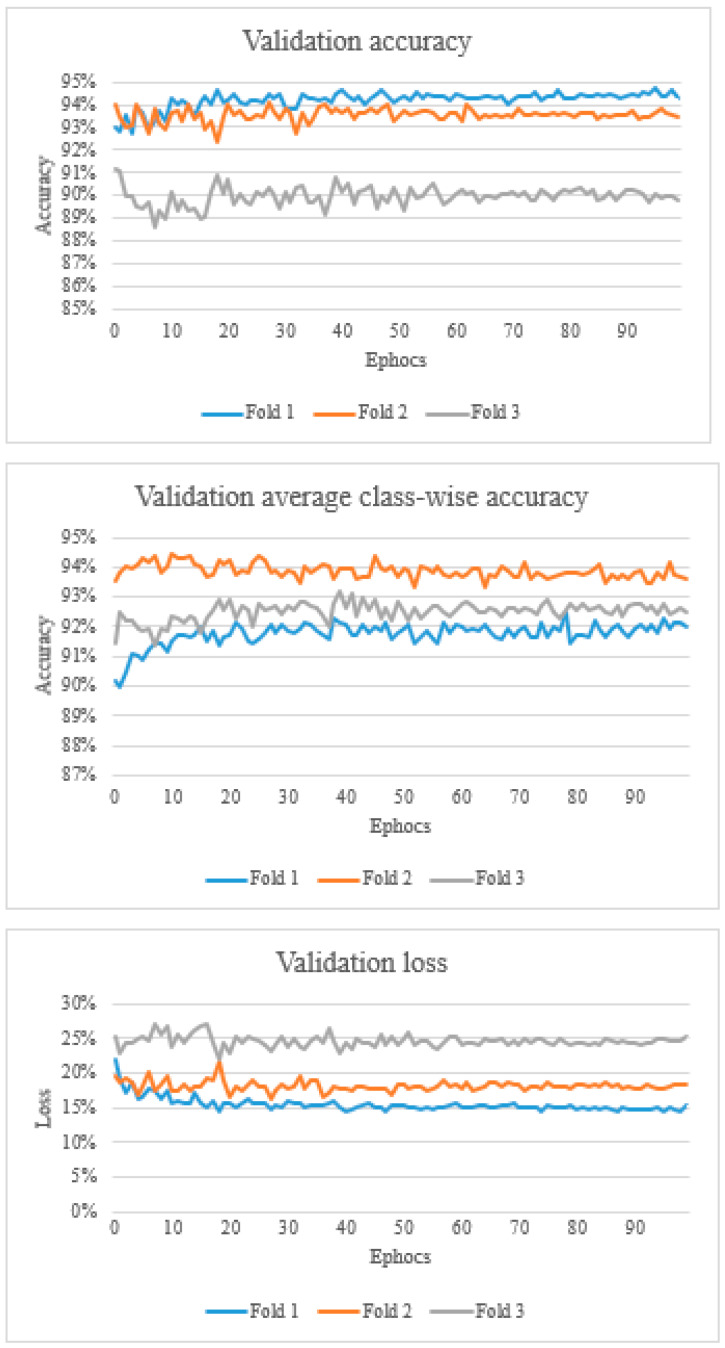
The top graph shows the validation accuracy, the middle graph shows the validation average class-wise accuracy, and the bottom graph shows the validation loss.

**Figure 4 vetsci-10-00045-f004:**
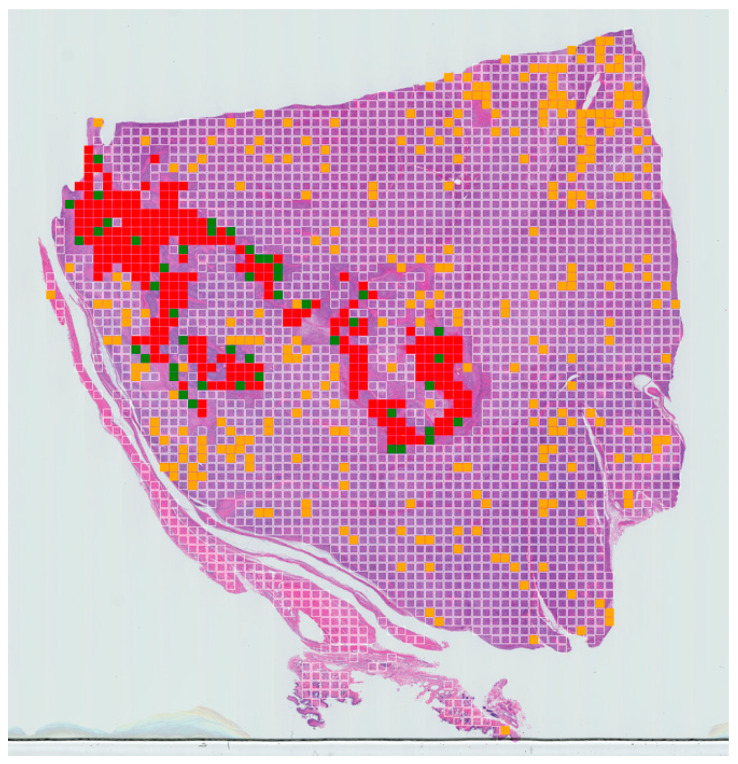
Example of slide-level confusion map showing areas of necrosis within a WSI of canine soft-tissue sarcoma. True positives (TPs) are displayed in red, false negatives (FNs) in green, false positives (FPs) in yellow and true negatives (TNs) in clear. These maps can be used to calculate the percentage of tumour necrosis in a patient and visualise the extent of the tumour necrosis over the whole-slide image.

**Table 1 vetsci-10-00045-t001:** The data in the table relate to information for the training and validation datasets for each experiment (folds 1, 2 and 3). Each table (folds 1, 2 and 3) contains information about WSI number, tumour grade, number of patches extracted from necrosis areas (positive patches) and normal tissue (negative patches) for each WSI. Each table also indicates the total number of positive and negative patches used for training and validation.

Slide Code	Grade	Fold 1	No. of Necrosis Patches 20×	No. of Negative Patches 20×
#1	1	Validation	0	2856
#2	1	Validation	0	3379
#3	2	Validation	0	2606
#4	2	Validation	314	1417
#5	3	Validation	86	2542
#6	3	Validation	1560	2324
**Total**			**1960**	**15,124**
#7	1	Training	0	800
#8	1	Training	0	800
#9	1	Training	0	800
#10	2	Training	41	800
#11	2	Training	0	800
#12	2	Training	0	800
#13	2	Training	0	800
#14	2	Training	1696	800
#15	2	Training	0	800
#16	3	Training	57	800
#17	3	Training	934	800
#18	3	Training	210	800
#19	3	Training	742	800
#20	3	Training	144	800
**Total**			**3824**	**11,200**
**Slide Code**	**Grade**	**Fold 2**	**No. of Necrosis** **Patches 20×**	**No. of Negative** **Patches 20×**
#9	1	Validation	0	2132
#8	1	Validation	0	2280
#11	2	Validation	0	4944
#19	3	Validation	742	2367
#17	3	Validation	934	1332
#20	3	Validation	144	1727
#18	3	Validation	210	2280
**Total**			**2030**	**17,062**
#1	1	Training	0	800
#2	1	Training	0	800
#7	1	Training	0	800
#4	2	Training	314	800
#10	2	Training	41	800
#4	2	Training	0	800
#12	2	Training	0	800
#13	2	Training	0	800
#14	2	Training	1696	800
#15	2	Training	0	800
#16	3	Training	57	800
#6	3	Training	1560	800
#5	3	Training	86	800
**Total**			**3754**	**10,400**
**Slide Code**	**Grade**	**Fold 3**	**No. of Necrosis** **Patches 20×**	**No. of Negative** **Patches 20×**
#7	1	Validation	0	2259
#12	2	Validation	0	3562
#13	2	Validation	0	2551
#14	2	Validation	1696	2119
#15	2	Validation	0	2936
#10	2	Validation	41	2983
#16	3	Validation	57	2379
**Total**			**1794**	**18,789**
#1	1	Training	0	800
#9	1	Training	0	800
#2	1	Training	0	800
#8	1	Training	0	800
#4	2	Training	314	800
#3	2	Training	0	800
#20	3	Training	144	800
#5	3	Training	86	800
#18	3	Training	210	800
#19	3	Training	742	800
#17	3	Training	934	800
#11	3	Training	0	800
#6	2	Training	1560	800
**Total**			**3990**	**10,400**

**Table 2 vetsci-10-00045-t002:** Test dataset for folds 1, 2 and 3. The data represented in the table contain information about slide code, tumour grade, number of patches containing necrosis (positive), number of positive and negative patches (total) and % of necrosis present in each WSI.

Slide Code	Grade	Positive	Total	Necrosis %
#21	1	0	4371	0.00
#22	1	0	1611	0.00
#23	1	0	2798	0.00
#24	1	0	3040	0.00
#25	2	14	1883	0.74
#26	2	0	3368	0.00
#27	2	20	1618	1.24
#28	2	2	2714	0.07
#29	3	302	3003	10.06
#30	3	138	4528	3.05
#31	3	369	3378	10.92
#32	3	306	2890	10.59

**Table 3 vetsci-10-00045-t003:** Accuracy, sensitivity precision and F1 score for validation and test for folds 1, 2 and 3.

Slide Code	Sensitivity/Recall (%)	Precision (%)	Accuracy (%)	F1-Score (%)
Fold1_validation	88.5	70.05	94.4	78.5
Fold1_test	93.4	30.0	92.7	45.4
Fold2_validation	94.3	63.2	93.6	75.7
Fold2_test	94.0	25.4	90.8	39.9
Fold3_validation	95.9	46.8	90.1	62.9
Fold3_test	94.6	22.4	89.1	36.2

**Table 4 vetsci-10-00045-t004:** Predicted positive labels and % of necrosis that the algorithm detected in each slide after training.

**Fold 1**			
**Slide**	**Predicted Positive**	**Total**	**Necrosis %**
#21	165	4371	3.77
#22	343	1883	18.22
#23	147	3040	4.84
#24	44	3368	1.31
#25	22	2798	0.79
#26	45	1618	2.78
#27	56	2714	2.06
#28	56	1611	3.48
#29	535	3003	17.82
#30	905	4528	19.99
#31	873	3378	25.84
#32	395	2890	13.67
**Fold 2**			
**Slide**	**Predicted Positive**	**Total**	**Necrosis %**
#21	192	4371	4.39
#22	343	1883	18.22
#23	270	3040	8.88
#24	61	3368	1.81
#25	31	2798	1.11
#26	53	1618	3.28
#27	55	2714	2.03
#28	76	1611	4.72
#29	537	3003	17.88
#30	1153	4528	25.46
#31	997	3378	29.51
#32	498	2890	17.23
**Fold 3**			
**Slide**	**Predicted Positive**	**Total**	**Necrosis %**
#21	388	4371	8.88
#22	412	1883	21.88
#23	474	3040	15.59
#24	70	3368	2.08
#25	30	2798	1.07
#26	46	1618	2.84
#27	74	2714	2.73
#28	76	1611	4.72
#29	621	3003	20.68
#30	1130	4528	24.96
#31	1172	3378	34.70
#32	368	2890	12.73

**Table 5 vetsci-10-00045-t005:** Predicted % of necrosis that the algorithm detected in each slide after training and the % of necrosis annotated by the pathologists.

Slide	Fold 1	Fold 2	Fold 3	Pathologists’ Annotations
#21	3.77	4.39	8.88	0.00
#22	18.22	18.22	21.88	0.00
#23	4.84	8.88	15.59	0.00
#24	1.31	1.81	2.08	0.00
#25	0.79	1.11	1.07	0.74
#26	2.78	3.28	2.84	0.00
#27	2.06	2.03	2.73	1.24
#28	3.48	4.72	4.72	0.07
#29	17.82	17.88	20.68	10.06
#30	19.99	25.46	24.96	3.05
#31	25.84	29.51	34.70	10.92
#32	13.67	17.23	12.73	10.59

## Data Availability

The datasets and algorithm code generated for this study are available on request to the corresponding author.

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
