# Peer review of "Detection of Necrosis in Digitised Whole-Slide Images for Better Grading of Canine Soft-Tissue Sarcomas Using Machine-Learning"

_vetsci, 2023, doi:10.3390/vetsci10010045_

Round 1
Reviewer 1 Report
The manuscript describes a study in which machine learning is used as a way to automate the detection of necrosis with STS. Much of the manuscript describes the methods used to develop the machine learning and I do not feel qualified to comment on the appropriateness of these methods.
My major uncertainty regarding the manuscript is that the authors state that the technique has an ‘accuracy’ of 92.7%. However, I couldn’t see how this was derived anywhere in the manuscript. Seven of the STS had necrosis – how was the 92.7% figure derived? Also, in lines 226-230 % of necrosis within a STS is given. How accurate was the algorithm in predicting the % of necrosis within each of the tumors? It would be good if this information was presented in a table with the % necrosis identified by the pathologist compared to the % necrosis identified by the computer analysis listed to allow easy comparison. Did the machine incorrectly identify necrosis in a STS which did not contain any on examination by a pathologist? This seems like the key finding and so this information should be presented as clearly as possible so readers are able to see how good the algorithm was at identifying and quantifying necrosis within a STS.
As before, both the simple summary and abstract should make it clear exactly what is being compared which allowed an accuracy of 92.7% to be stated.
The introduction as written is just one very long paragraph – I think if this is re-organized, this could improve readability. Potentially some of the other studies when this technique was used in human neoplasms could be shortened – maybe just focus on the ones that use this to detect necrosis.
Line 106. The grade isn’t really important for this study – it is all about detecting and quantifying necrosis.
Line 226. How the % of necrosis was determined is not described in the methods. Likewise line 231 seems to be methods not results.
Table 3 needs a legend to define abbreviations and some of this table seems to be methods rather than results.
The discussion could be improved as it currently has a number of single sentence paragraphs. It also refers to tables and figures – this should only be done in results. For Discussion they reader should know what results you have presented – you are discussing how they are to be interpreted not reviewing them. In the discussion I would like to see more information about how this would be applied. As mentioned in the last paragraph, pathologists do not have difficulties detecting necrosis within a sample. However, the variability is more important in the trimming of the tumors and often necrotic areas of tumor are deliberately not trimmed for histological examination. Would the variability be better solved if trimming was standardized rather than by developing computer learning?
Reviewer 2 Report
The auhors descirbe the automatic assessment of whole slide images for the detection and quantification of necrosis in canine soft tissue sarcoma. The manuscript is well written, the data is well presented and discussed. Abstract is concise and clearly written, and clear representation of the aim of the paper and the title adequately reflects the subject under investigation. Materials and Methods described the sampling procedure, sample preparation, analysis parameters, mutlivariate data analysis and machine learning. I would recommend this publication as is.
Author Response
Dear reviewer,
Thank you very much for approving our manuscript.
Kind regards,
Ambra